# Tyrosine Kinase Inhibitors Are Promising Therapeutic Tools for Cats with HER2-Positive Mammary Carcinoma

**DOI:** 10.3390/pharmaceutics13030346

**Published:** 2021-03-06

**Authors:** Andreia Gameiro, Filipe Almeida, Catarina Nascimento, Jorge Correia, Fernando Ferreira

**Affiliations:** 1CIISA—Centro de Investigação Interdisciplinar em Sanidade Animal, Faculdade de Medicina Veterinária, Universidade de Lisboa, Avenida da Universidade Técnica, 1300-477 Lisboa, Portugal; agameiro@fmv.ulisboa.pt (A.G.); filipe.almeida@insa.min-saude.pt (F.A.); catnasc@fmv.ulisboa.pt (C.N.); jcorreia@fmv.ulisboa.pt (J.C.); 2Antiviral Resistance Laboratory, Infectious Diseases Department, National Institute of Health Dr. Ricardo Jorge, Av. Padre Cruz, 1649-016 Lisbon, Portugal

**Keywords:** feline mammary carcinoma, HER2, tyrosine kinase inhibitors, targeted therapies, feline *her2* TK mutations

## Abstract

Feline mammary carcinoma (FMC) is a common neoplasia in cat, being HER2-positive the most prevalent subtype. In woman’s breast cancer, tyrosine kinase inhibitors (TKi) are used as a therapeutic option, by blocking the phosphorylation of the HER2 tyrosine kinase domain. Moreover, clinical trials demonstrated that TKi produce synergistic antiproliferative effects in combination with mTOR inhibitors, overcoming resistance to therapy. Thus, to uncover new chemotherapeutic strategies for cats, the antiproliferative effects of two TKi (lapatinib and neratinib), and their combination with a mTOR inhibitor (rapamycin), were evaluated in FMC cell lines (CAT-M, FMCp and FMCm) and compared with a human breast cancer cell line (SkBR-3). Results revealed that both TKi induced antiproliferative effects in all feline cell lines, by blocking the phosphorylation of EGFR members and its downstream effectors. Furthermore, combined treatments with rapamycin presented synergetic antiproliferative effects. Additionally, the DNA sequence of the *her2* TK domain (exons 18 to 20) was determined in 40 FMC tissue samples, and despite several mutations were found none of them were described as inducing resistance to therapy. Altogether, our results demonstrated that TKi and combined protocols may be useful in the treatment of cats with mammary carcinomas, and that TKi-resistant FMC are rare.

## 1. Introduction

The feline mammary carcinoma (FMC) is one of the most common feline tumors (12% to 40% of all neoplasms) [1,2], sharing similar clinicopathological features and histologic subtypes with human breast cancer [3], making the cat a suitable model for comparative oncology studies [4,5,6,7]. Indeed, cats with HER2-positive mammary carcinoma also show a poor prognosis, with high clinical tumor aggressiveness, metastization capability and shorter overall-survival (OS) [1,2,8,9], with the overexpression of the epidermal growth factor receptor 2 (HER2) been reported in a range from 33% to 60% of all FMC cases [10]. Unfortunately, no specific therapeutic targets are available for cats with HER2-positive mammary tumors, being the adjuvant chemotherapy unbeneficial in most of the cases, with the radical surgery being the unique therapeutic strategy [11]. Thus, new molecular targets are needed to develop more efficient therapeutic protocols.

In cat, the *her2* gene is localized in the chromosome E1, presenting a sequence identity of 92% with its human counterpart [12,13]. The HER2 is a tyrosine kinase (TK) transmembrane glycoprotein (186 kDa), constituted by three domains: an extracellular region, a transmembrane domain and an intracellular subdomain with TK activity [14]. In cell membrane, the HER2 can dimerize with itself or with other members of the epidermal growth factor receptor (EGFR) family, showing a proto-oncogene activity, with the ability to modulate cell cycle and apoptotic responses [1,12,14]. Furthermore, HER2 phosphorylation promotes cell proliferation via the RAS-ERK pathway and inhibits cell death via the PI3K-AKT-mTOR pathway, which makes the phosphorylated HER2 a marker of poor prognosis [15] and a useful molecular target in breast cancer therapy. However 50% of patients with HER2-positive breast cancer show immunotherapy resistance to anti-HER2 monoclonal antibodies, being the use of TK inhibitors (TKi) an alternative approach for these targeted immunotherapies. The TKi are small chemical compounds that prevent HER2 phosphorylation, by competing with the adenosine triphosphate (ATP) at the cytoplasmic catalytic kinase domain and, consequently, blocking the signaling for the downstream cascade activation [16]. Several TKi are conventionally used in human breast cancer therapy, such as lapatinib and neratinib. Accordingly, lapatinib is a reversible dual EGFR inhibitor, binding the ATP-binding pocket of HER1 and HER2 proteins [17], presenting apoptotic effects [16]. This compound has a specific binding location, encoded by exon 20 of the *her2* gene, with several point mutations being associated with therapy resistance (e.g., L755S, T798M) [18]. In parallel, neratinib is an irreversible pan-HER inhibitor that interacts with HER1, HER2 and HER4 proteins [14], able to overcome lapatinib therapy resistance [18]. This molecule binds the TK ATP binding pocket, with a high binding specificity in a conserved cysteine (Cys) residue [19], leading to cell death, by a ferroptosis mechanism [20,21], with woman breast cancer patients having the acquired mutation at the residue T798I in HER2 and/or HER4 overexpression showing neratinib resistance [22]. Indeed, the HER4 protein is crucial for normal mammalian gland development and differentiation, and it is hypothesized that HER2-positive tumor cells can shift their dependency from HER2 to HER4 [23], to maintain cell survival and growth.

Nowadays, the combined treatments, such as the use of TKi conjugated with the mammalian target of rapamycin (mTOR) inhibitor, are valuable therapeutic tools, in order to reduce resistance development and drug secondary effects. Indeed, the rapamycin that is a fungicide macrolide with immunosuppressive and anticancer properties, binds to an immunophilin protein family, inhibiting the serine-threonine kinase mTOR [24,25] and its complex (mTORC) involved in cell cycle progression and cell proliferation [25], two hallmarks of the carcinogenesis process [24].

For breast cancer patients, treatments based on precision and target specificity are crucial towards the improvement of the therapeutic response and preventing drug resistance [17]. Once no data is available about the use of these compounds in feline mammary carcinoma, this study aims to: (1) evaluate the antiproliferative effects of two TKi (lapatinib and neratinib) using three feline mammary carcinoma cell lines (CAT-M, FMCp and FMCm); (2) characterize the effect of the TKi in the phosphorylation patterns on EGFR family members (HER1, HER2 and HER4), its downstream pathways (PI3K/AKT and ERK1/2) and on the tumor suppressor PTEN protein; (3) measure the cytotoxic effects of the mTOR inhibitor, rapamycin and evaluate its effect on the mTOR phosphorylation pattern; (4) evaluate the synergistic antiproliferative effects of the combined treatments of using TKi and rapamycin; and (5) identify genomic mutations in the feline *her2* gene TK domain in FMC tissue samples, in order to assess their use as prognostic/drug-resistance biomarkers.

## 2. Materials and Methods

### 2.1. Feline and Human Mammary Carcinoma Cell Lines

In this study, three feline mammary carcinoma cell lines (CAT-M, from the American Type Culture Collection (ATCC), Manassas, VA, USA; FMCp and FMCm kindly provided by Prof. Nobuo Sasaki (DVM) and Prof. Takayuki Nakagawa (DVM), University of Tokyo, Japan) were immunophenotyped and tested, as well as a HER2-overexpressing human breast cancer cell line (SkBR-3, from the ATCC) used as positive control (Appendix A). CAT-M and SkBR-3 cell lines were maintained in Dulbecco’s modified Eagle Medium (DMEM; Corning, New York, NY, USA), whereas FMCp and FMCm cell lines were growing in Roswell Park Memorial Institute 1640 Medium (RPMI; Corning), both supplemented with heat-inactivated 20% (*v*/*v*) fetal bovine serum (FBS; Corning), and incubated at 37 °C, in a humidified atmosphere of 5% (*v*/*v*) CO_2_ (Nuaire, Plymouth, MN, USA). Periodically, cell lines were tested for Mycoplasma (MycoSEQ™ Mycoplasma Detection Kit, Thermo Fisher, Waltham, MA, USA) and examined for their morphology and proliferation rate.

### 2.2. In Vitro Cytotoxicity Assays

To determine the effect of the TKi (lapatinib and neratinib; Sigma-Aldrich, Darmstadt, Germany) and of the mTOR inhibitor (rapamycin; Sigma-Aldrich), viability assays were performed using the Cell Proliferation Reagent WST-1 kit (abcam, Cambridge, UK), following to the manufacturer’s instructions. Briefly, cell lines were seeded in 96-well plates to reach a 90% confluency at 24 h. Then, cells were exposed to increasing concentrations of each drug to obtain a dose-response plot (Table 1), while control wells were left unexposed. Dimethyl sulfoxide (DMSO; Sigma-Aldrich) was used as vehicle control, at a maximum final concentration of 1%. After 72 h of drug exposure, the WST-1 reagent (abcam) was added during an incubation period of 4 h, at 37 °C, and absorbance at 440 nm was measured using a plate reader (FLUOStar Optima, BMG LabTech, GmbH, Offenburg, Germany). Triplicate wells were used to determine each data point and three independent experiments were performed. 

In combined treatments (lapatinib plus rapamycin; neratinib plus rapamycin) the same strategy was followed, with the concentrations chosen in a range that a small or no cytotoxic effect was observed.

### 2.3. Immunoblotting

For the quantification of protein expression levels, the western blot technique was performed as previously reported by us [2,26], and other groups [27,28]. Briefly, for the preparation of whole cell extracts, cell lines were seeded in 6-well plates, in order to obtain a confluency of 90%, after 24 h. Then, cells were exposed to drugs, using IC_50_ concentrations, and a negative control was left unexposed, using DMSO (Sigma-Aldrich) as a vehicle. After 72 h of exposure, cells were collected using 200µL of RIPA lysis buffer (50 mM TrisHCl, 150 mM NaCl, 1% Tween-20, 0.5% sodium deoxycholate, 0.1% SDS, 5 mM EDTA, pH 8.0) supplemented with 1% 100X Halt™ protease inhibitor cocktail EDTA-free (Thermo Fischer) and 2 mg/mL iodoacetamide (AppliChem, GmbH, Darmstadt, Germany). Afterwards, whole cell extracts were resuspended with 10 µL of SDS-PAGE Loading Buffer 5X (NZYTech, Lisbon, Portugal), denatured for 10 min at 96 °C, and stored at −80 °C, until further use. Before immunoblotting, the extracts were fully thawed, and then resolved in 7.5% SDS-PAGE. Proteins were transferred onto nitrocellulose membranes (Amersham Protran 0.45 NC, GE Healthcare, Chicago, IL, USA) and blocked in 5% (*w*/*v*) dried skimmed milk diluted in phosphate buffered saline (PBS)−0.1% (*v*/*v*) Tween-20 (National Diagnostics, Atlanta, GA, USA). Membranes were probed with primary antibodies (abcam; Table 2) diluted in 1% (*w*/*v*) dried skimmed milk in PBS−0.1% Tween-20 (National Diagnostics), overnight at 4 °C, and then with horseradish peroxidase (HRP)-conjugated secondary antibodies (Goat HRP anti-rabbit IgG H&L and goat HRP anti-mouse IgG H&L, 1:10,000, Abcam) for 1 h at room temperature (RT). Bands were detected using ECL (Clarity Substrate, Bio-Rad, Hercules, CA, USA), visualized and analyzed using a Chemidoc XRS + system with Image Lab capture software (Bio-Rad).

### 2.4. Animal Population

The forty cats diagnosed with mammary carcinoma enrolled in this study, were recruited at the Hospital of Faculty of Veterinary Medicine, University of Lisbon, being the surgical procedures consented by the owners, and without interfering with the animals’ well-being. Their clinical history was fully documented (Table 3) [2], including breed, age, reproductive status and contraceptive administration, treatment (surgery, or surgery plus chemotherapy), number, location and size of tumor lesions, histopathological classification, tumor immunophenotype [3,29], malignancy grade [30], presence of tumor necrosis, lymphatic invasion, lymphocytic infiltration, cutaneous ulceration, regional lymph node involvement and clinical stage (TNM system). All tissue samples were frozen at −80 °C and stored until further use.

### 2.5. DNA Extraction, Amplification and Sequence Analysis of the Feline her2 TK Domain

Genomic DNA extraction was performed in the three feline cell lines, collected after reaching confluence in a T25 culture flask, and in 5 mg of 44 frozen tissue samples (40 feline mammary tumor samples and 4 breed controls), as previously described [31,32,33], using a QIAmp FFPE kit (Qiagen, Dusseldorf, Germany) and following the manufacturer’s recommendations. Tissue samples were homogenized with the Tissue Lyser II (Qiagen), and all samples were digested with protease K (20 mg/mL; Qiagen). After several washing steps, the genomic DNA was eluted from the extraction columns and its quality and quantity was measured in NanoDrop ND-100 Spectrophotometer (Thermo Fisher Scientific). The feline *her2* TK domain (NC_018736.3) was identified by comparison with the genomic human *her2* sequence (NC_000017.11), and primers for DNA amplification of exons 18 to 22 were designed in the Primer designing tool (NCBI, Bethesda, MD, USA) (Table 4). PCR technique was performed with a standard reaction mixture (4 µL/sample of Phusion GC buffer from Thermo Fischer Scientific; 0.4 µL/sample of dNTPs from Grisp, Porto, Portugal; 0.1 µL/sample of each forward and reverse primer and 0.2 µL/sample of DNA polymerase from Thermo Fischer Scientific) at a final DNA concentration of 4 ng/mL. For amplification of the exons 18 and 19 of the *her2* gene, PCR reactions were performed in a PCR Thermal Cycler (VWR, Leicestershire, England) as follows: denaturation at 98 °C for 30 s, followed by 35 cycles at 98 °C for 10 s, 60 °C for 30 s, 72 °C for 10 s, plus one final extension step at 72 °C for 10 min. For exons 20 to 22, the melting temperature was 58 °C. After confirmation of the expected size for each amplified sequence in a 2% agarose gel (Sigma-Aldrich), DNA fragments were purified and sequenced by Sanger technique (StabVida, Almada, Portugal). DNA sequences were inspected for inaccuracies.

The feline sequenced samples were aligned with the identified feline *her2* (NC_018736.3), using the ClustalW tool (BioEdit Alignment Editor software) and the consensus sequence was confirmed using SeqTrace 9.1 software [34], while SNP and protein amino acid changes were identified by using Expert Protein Analysis System (ExPASY) translate tool and compared with the original protein sequence (NP_001041628.1, NCBI). Mutations identified in the feline tissue samples were compared to the human *her2* sequence (NC_000017.11) and searched in National Cancer Institute database, International Cancer Genome Consortium and Catalogue of Somatic Mutations in Cancer (COSMIC) databases for putatively induced resistance to the TKi tested in this study.

### 2.6. Statistical Analysis

Statistical analysis was carried out using the GraphPad Prism software (version 5.04, for Windows, San Diego, CA, USA), with two-tailed *p*-values less than 0.05 being considered statistically significant, and a 95% confidence interval (* *p* < 0.05, ** *p* < 0.01 and *** *p* < 0.001). Regarding the cytotoxicity assays, outliers with more than two standard deviations were removed and the IC_50_ value for each drug was calculated using Log (Inhibitor) vs. Response (Variable slope) function. For the drug conjugation assays, the two-way ANOVA test was performed. In the animal population, correlations considering the genomic mutations were assessed between groups using the non-parametric Mann Whitney test.

## 3. Results

### 3.1. Lapatinib and Neratinib Exert Antiproliferative Effects in All Feline Mammary Carcinoma Cell Lines

Results showed that incubation of FMC cell lines with lapatinib or neratinib, exert potent antiproliferative effects in a dose-dependent fashion. Indeed, the lapatinib was able to induce 100% of cytotoxicity in all feline cell lines (CAT-M with a IC_50_ = 3930 nM ± 49, Figure 1A; FMCp with a IC_50_ = 4870 nM ± 100, Figure 1B; FMCm with a IC_50_ = 17,470 nM ± 100, Figure 1C), while neratinib showed lower cytotoxicity (33.5% of cytotoxicity in CAT-M cells, Figure 1A; 79.4% of cytotoxicity in FMCp cells, Figure 1B; and 31.4% of cytotoxicity in FMCm, Figure 1C). As expected, the human breast cancer SkBR-3 cell line showed 91.1% of cytotoxicity when incubated with lapatinib (IC_50_ = 16,220 nM ± 1040) and a cytotoxicity of 60.5% in the presence of neratinib (Figure 1D).

### 3.2. Lapatinib and Neratinib Inhibit the Phosphorylation of HER2 and Its Downstream Cascade

Considering the promising results obtained from the incubation of all feline cell lines to both TKi, the HER2 expression was evaluated by immunoblot analysis. Results revealing that HER2 expression is much lower in feline cell lines than in human positive control (SkBR-3), being the cell line FMCp HER2-negative (Figure 2A). Once the HER2 expression in feline cell lines was higher in CAT-M cells, further analysis was performed to characterize the effects of TKi on the HER2 signaling pathway activity using this cell line. After lapatinib exposure, a decreasing on the phosphorylation levels of HER1 and HER2 was detected, coupled with a reduction in the phosphorylation levels of the downstream effectors AKT and ERK1/2. Moreover, an increase in phosphorylated PTEN at Thr366 was also observed (Figure 2B). Corroborating the lower cytotoxicity induced by neratinib incubations, differences in the phosphorylated levels were less noticeable (Figure 2B).

Interestingly, when feline tumor cells were incubated with TKi, in particularly, with lapatinib, the HER2 expression levels increased, both in CAT-M (Figure 2C) and FMCp (Figure 2D) cell lines.

### 3.3. Rapamycin Does Not Induce Strong Antiproliferative Effects in Feline Cell Lines

The use of rapamycin as a single drug did not presented a valuable antiproliferative effect (Figure 3A). Accordingly, the CAT-M and FMCm cell lines showed 38.7% and 41.3% of cytotoxicity, respectively, with the FMCp showing slightly more sensitivity to rapamycin (43.3% of cytotoxicity). Similar results were obtained with the control SkBR-3 cell line (26.9% of cytotoxicity). Additionally, the immunoblot analysis revealed a decrease in the mTOR phosphorylation levels after rapamycin exposure (Figure 3B).

### 3.4. Combined Exposures of TKi and mTOR Inhibitor Showed Strong Synergistic Antiproliferative Effects

The combined exposures of lapatinib plus rapamycin and neratinib plus rapamycin, demonstrated high synergistic cytotoxic effects regardless of feline cell lines. For instance, in the CAT-M cell line, the cytotoxic effects increased from 5.6% to 57.5% (*p* = 0.0360) by adding 6.25 nM of rapamycin to 3125 nM of lapatinib and raised from 2.5% to 49.9% (*p* = 0.0044) when cells were incubated with 12.5 nM of neratinib plus 6.25 nM of rapamycin (Figure 4A). Similarly, in FMCp cell line, the antiproliferative effects increased from 6.9% to 54.4% (*p* < 0.001) when lapatinib at 780 nM was combined with rapamycin at 6.25 nM and from 0.4% to 44.5% (*p* = 0.0034) when neratinib at 3.125 nM was combined with rapamycin at 6.25 nM (Figure 4B). In the metastatic FMCm cell line, the higher antiproliferative effects were obtained by the use of lapatinib at 22,650 nM plus rapamycin at 6.25 nM, increasing from 9.6% to 95.2% (*p* < 0.001), and exposing cells to neratinib at 25 nM plus rapamycin at 6.25 nM, increasing the antiproliferative effects from 9.6% to 76.3% (*p* < 0.001; Figure 4C). In the control SkBR-3 cell line, also synergistic effects were detected in all combination assays (Figure 4D).

### 3.5. Mutations Found in the her2 TK Domain of FMC Clinical Samples Were Not Associated with TKi Therapy Resistance

Previous studies on somatic mutations in the TK domain of the *her2* gene, in human breast cancer patients, revealed several mutations associated with therapy resistance to TKi and/or specific clinicopathological features, reported in National Cancer Institute database, International Cancer Genome Consortium and COSMIC databases. In this work, the mutational analysis of the feline *her2* TK domain, showed that mutations occur in the majority of tumor samples (90%, 36/40), being identified a total of 42 single variants (SVs) (Figure 5), located in introns (54.8%; 23/42) and exons (45.2%; 19/42) (Appendix A). Regarding the intronic SVs, 21.7% (5/23) were found in splicing regions (c.19631 and c.19643; c.20278 and c.20289; c.20612), with the majority detected between exons 18 and 19 (60.9%; 14/23). Further sequence analysis revealed that exons 18 and 20 showed a higher number of mutations (57.9%, 11/19 and 36.8%, 7/19, respectively), contrasting with the exon 22 that has only one mutation with a very high frequency (85%; 34/40; c.20940 T > G) and with exons 19 and 21 where no mutations were detected. Finally, when tumor samples were divided into four molecular subtypes, 71.4% of the triple-negative tumors (5/7), 58.3% of the HER2-positive (7/12) and 55.6% of the luminal B (10/18) showed at least one mutation. No mutations were detected in luminal A tumors (0/3). 

A special consideration was made at exon 20, which encodes for the local binding of the TKi. In this exon 7 mutations were reported, in a total of 13 samples, and occurring 46.2% (6/13) in the HER2-positive tumor subtype. Furthermore, evaluating the SVs, 4 missense mutations (c.20380 C > G; c.20384 A > T; c.20428 G > C; c.20459 A > T), 2 synonymous mutations (c.20382 T > C; 20436 G > A) and 1 base change that leads to a STOP codon (c.20385 T > G) were observed. 

After comparing the feline mutations with the genomic human *her2* sequence, and searching in the referred databases, none of the mutations were reported as inducing resistance to TKi tested.

In order to identify new prognostic biomarkers, correlations were made between the SVs reported and animals’ clinicopathological features. This analysis did not reveal any correlation, beyond the mutation c.19573A > T, located in exon 18 associated with tumor size (* *p* = 0.045; Figure 6), revealing that the mean rank of the general samples was 2.64 cm ± 1.13, comparing with 4.13 cm ± 1.44, of the samples presenting that mutation.

## 4. Discussion

Feline mammary carcinoma is usually diagnosed belatedly, presenting an aggressive behavior, with the FMC HER2-positive showing a poor prognosis [3,8,35]. Therefore, an early detection and an effective therapy is crucial, in order to improve survival time in cats with mammary tumors. Thus, in this study, the antiproliferative effects of two TKi (lapatinib and neratinib) were evaluated using three FMC cell lines (CAT-M, FMCp and FMCm), with different HER2 expression levels and no mutations described as possibly conducing to resistance to therapy (Appendix A). Moreover, the obtained results were compared to a human HER2-overexpressing breast cancer cell line (SkBR-3).

In therapeutic protocols, TKi inhibit the phosphorylation of EGFR family members, blocking its downstream pathways, such as lapatinib. This TKi was approved for solid, and metastatic HER2-positive breast tumors [36,37] preventing the phosphorylation of HER1 (T1173) and HER2 (T1221/1222), markers of poor prognosis in breast cancer patients [15,23]. Lapatinib was reported as inducing an antiproliferative effect in cells expressing different HER2 concentrations [38], as proved in this assay, being the HER2-positive CAT-M cells the most susceptible (IC_50_ = 3930 nM ± 49), with similar results obtained in the SkBR-3 cell line (IC_50_ = 16,220 nM ± 1040) [39,40,41,42]. Although, 100% of cytotoxicity was also observed in the HER2-negative FMCp cell line (IC_50_ = 4870 nM ± 100). Indeed, studies in humans demonstrated that lapatinib is useful in patients with triple-negative breast tumors, by activating NF-kB and the anti-apoptotic Bcl-2 protein [43], sensitizing tumor cells to the annexin A6 upregulation [44] and inducing apoptosis [45]. Furthermore, lapatinib also interacts with HER1, which is usually upregulated in triple-negative tumors, being suggested as an alternative therapeutic tool for patients with this breast cancer subtype [46,47]. In parallel, the metastatic cell line (FMCm) showed the highest IC_50_ value (17,470 nM ± 100), suggesting that in metastatic tumors, lapatinib is more useful when combined with other compounds [36,37]. In addition, the molecular effects of lapatinib seem to be conserved between human and feline cell lines, with CAT-M cells showing lower phosphorylation levels of HER2 (Y1221 + Y1222), HER1 (Y1173), AKT (S473) [41,48] and ERK1/2 (T202/Y204 + T185/Y187) [40,49,50,51,52] after lapatinib exposure. Results obtained also demonstrated that lapatinib reduced the phosphorylation levels of PTEN at T366, putatively through the glycogen synthase kinase 3 (GSK-3), playing a role in the destabilization of the protein, which leads to a more potent effect in the suppression of cancer cells proliferation [53]. Additionally, as previously reported in humans, lapatinib also increased the HER2 expression levels, possibly by stabilizing the protein at the cell membrane and inhibiting its ubiquitination [39]. Interestingly, this effect was also observed in the HER2-negative FMCp cell line, which could support the antiproliferative effects obtained by lapatinib. 

The other TKi tested in this study was neratinib, a compound approved for the adjuvant treatment of early-stage and metastatic HER2-positive breast cancer (Food and Drug Administration). Interestingly, this TKi interacts with different EGFR family members (HER1, HER2 and HER4), inducing ubiquitination and lysosomal degradation of the HER2 [54], and it is able to reverse membrane-bound ATP transporters surpassing multidrug resistance [55]. In the neratinib assay cytotoxic effects were reported for all the FMC cell lines. Indeed, CAT-M cell line presented a maximum of 33.5% of cytotoxicity, smaller when compared to the human, SkBR-3 cell line (60.5% of cytotoxicity) [54], possibly due to CAT-M cells’ lower HER2 expression levels. Notably, the FMCp cell line presented a high sensitivity to neratinib (74.9% of cytotoxicity), suggesting its effect in other EGFR family members [20,56], and supporting the use of neratinib in HER2-negative breast cancer [57]. On the other hand, the metastatic FMCm cell line, revealed to be the less susceptible to neratinib (31.4% of cytotoxicity), suggesting its use as adjuvant in metastatic tumors [58,59]. Furthermore, in this cell line, the use of neratinib did not demonstrate a dose-dependent effect. This kind of behavior is documented for humans, e.g., by NmU overexpression [60], a protein involved in breast cancer progression and metastization [61], or by an increased activity of the enzyme cytochrome P4503A4 [62], leading to resistance to therapy. Additionally, after exposing cells to neratinib, it was not possible to demonstrate its molecular effects. In fact, only a slightly decrease in the phosphorylation levels of HER1 (Y1173) [63] were observed. Moreover, an increase in HER2 was obtained by exposure to neratinib, suggesting its stabilization at the cell membrane, by preventing protein phosphorylation.

Another valuable target in tumor therapy is the mTOR pathway, described as being hyperactivated in 80% of human cancers [64], and particularly in cats mTOR is involved in metastization, invasion and tumor progression [65]. The use of a mTOR inhibitor, rapamycin, in FMC cell lines revealed weak antiproliferative effects, similarly to the human SkBR-3 cell line, and as reported in humans [24,64,66]. However, the highest cytotoxic effect was obtained for the FMCp cell line (43.3% of cytotoxicity), corroborating the data that mTOR is frequently overexpressed in HER2-negative breast cancer [67]. Moreover, our results showed that the cytotoxicity of rapamycin did not demonstrate a dose-dependent effect, suggesting that cells were able to do the efflux of rapamycin, as reported in humans, by the overexpression of the ABCB1 transporter [68]. Additionally, evaluating the mTOR (S2448), using the CAT-M cell line as example, a decrease in its phosphorylation pattern was observed after exposure to rapamycin, as documented for women [24,69,70]. Since in breast cancer therapy several advantages are known to result from the combined treatments [19,71], and after the characterization of all the above compounds a synergistic effect of its combinations were demonstrated in the feline mammary carcinoma cell lines. The major advantage of this combined protocols is to block simultaneously different cell pathways, and overcome acquired resistant patterns, as documented for lapatinib, e.g., by HER3 [24] or HER4 activated pathways [23], cell signal reprogramming [72], recovering of the AKT/mTOR pathway [73], or Src-dependent resistance [40]. Furthermore, also for the TKi neratinib, acquired resistance is described, e.g., by NmU overexpression [60], or the reactivation of the proto-oncogene YES-1, a member of the Src family [74]. In the combined protocols, the best results were achieved in the FMCm cell line, with the highest increase in the cytotoxic effect occurring by the combination of lapatinib at 22,650 nM plus rapamycin at 6.25 nM, with an increase of 85.6% of cytotoxicity (from 9.6% to 95.2%), and for the combination of neratinib at 25 nM plus rapamycin at 6.25 nM, where an increase of 66.7% of cytotoxicity (from 9.6% to 76.3%) was obtained. The reported results, suggest the use of combined protocols as a valuable tool for feline mammary tumor therapy, as described for women breast cancer patients [19,71], particularly in metastatic [36,37] and triple-negative tumors [71]. Indeed, for the FMCp cell line, HER2-negative, an increase in 47.5% of cytotoxicity (from 6.9% to 54.4%) was obtained in the conjugation of lapatinib at 780 nM plus rapamycin at 6.25 nM, and by the use of neratinib at 3.125 nM plus rapamycin at 6.25 nM, where an increase of 44.1% of cytotoxicity (from 0.4% to 44.5%) was observed.

Beyond the referred acquired resistance to TKi, genomic mutations could also be responsible for a resistant profile to lapatinib, as well as neratinib. In breast cancer patients is known that *her2* is mutated in 2 to 3% of primary tumors and more than 70% of the mutations occurs in HER2-negative breast cancer subtype [75], as in the FMC tumor tissue samples. In this study, a total of 42 different SVs were found in the feline *her2*-TK domain, with 90% (36/40) of the clinical samples showing at least one mutation. Furthermore, mutations revealed to be more common in intronic regions (54.8%; 23/42), particularly, between the exon 18 and 19 (60.9%), with 21.7% of them being identified in the splicing regions. The intron splicing locations revealed to be of extreme importance [76], as already reported in *her2* gene in cats [77] and in women, being associated with breast cancer risk [78] and responsible for resistance to therapy [79,80]. Regarding the exons, several mutations were found (45.2%; 19/42), with the majority of them showing a low frequency. Most of the mutations identified were located at the exon 18 (57.9%), being the heterozygous synonymous mutation c.19573 A > C associated to larger tumor sizes (*p* = 0.044) and occurring in luminal B and triple-negative carcinoma subtypes. Considering exon 20, which encodes for the location recognized by both TKi [18,19], it comprises 36.8% of the mutations, in the FMC clinical samples. In the case of breast cancer patients mutations within this exon, they were reported to increase the HER2 catalytic activity [81,82], being the L755 (e.g., L755S and L755P), associated with resistance to lapatinib therapy [58,81,83,84], and L869R, together with T798I, promoting HER2 signaling and oncogenic growth, associated to neratinib acquired resistance [22,58]. Moreover, a rare mutation in breast tumors was identified in the feline luminal B and HER2-positive tumor subtypes, c.20385 T > G, that encodes for a STOP codon, leading to a truncated form of the protein and being associated with therapeutic resistance [85]. Furthermore, the mutations described should not compromise the use of TKi in cats, since none of them were described as inducing resistance to therapy in breast cancer patients, according to National Cancer Institute database, International Cancer Genome Consortium and COSMIC databases. 

## 5. Conclusions

Beyond the cat being considered a noble breast cancer model, in this study it was possible to demonstrate valuable antiproliferative effects, and a conserved action mechanism of lapatinib and neratinib in three distinct feline mammary carcinoma cell lines, similarly to the results reported in the human breast cancer cell line (SkBR-3). Moreover, the obtained data suggests the use of FMC cell lines as in vitro tools for screening of new therapeutic drugs. In this study, the best results were achieved by the use of lapatinb, being possible to demonstrate a decrease in the phosphorylation pattern of the EGFR family members and its downstream pathways. Furthermore, combined assays with TKi and mTOR inhibitor showed synergistic effects, anticipating the benefits of using small drug doses. Since mutations found suggest that TKi-resistant FMC are very rare, altogether, our in vitro results strongly suggest the potential usefulness of TKi alone or combined with rapamycin in the treatment of cats with feline mammary carcinoma.

## Figures and Tables

**Figure 1 pharmaceutics-13-00346-f001:**
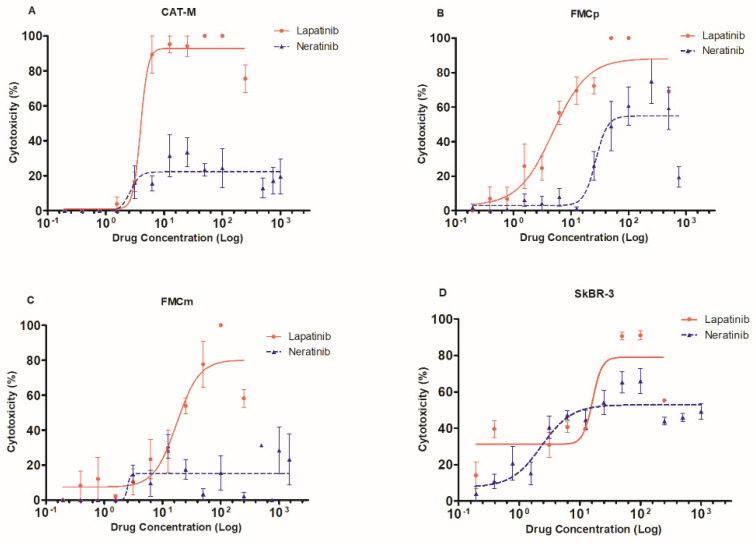
Lapatinib and neratinib showed strong cytotoxic effects in feline mammary carcinoma cell lines. (**A**) CAT-M cell line presented higher cytotoxic effects when incubated with lapatinib (IC_50_ = 3930 nM ± 49) than with neratinib. (**B**) FMCp cell line was susceptible to both lapatinib and neratinib (100% and 74.9% of cytotoxicity, respectively; IC_50_ = 4870 nM ± 100 for lapatinib). (**C**) FMCm cell line also showed high cytotoxic effects in the presence of lapatinib (IC_50_ = 17,470 nM ± 100), contrasting with the neratinib. (**D**) SkBR-3 cells showed a maximum cytotoxic effect of 91.1% (IC_50_ = 16,220 nM ± 1040) after exposure to lapatinib and 66.0% when exposed to neratinib. For graphical convenience, lapatinib was represented in a µM range, while neratinib was defined in a nM range. The experiments were performed in triplicates, in three independent assays.

**Figure 2 pharmaceutics-13-00346-f002:**
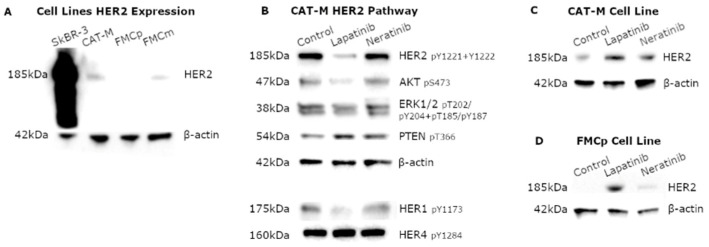
Feline cell lines presented low HER2 expression, showing altered phosphorylation levels of HER2 and its downstream effectors after TKi exposure. (**A**) CAT-M and FMCm cell lines presented low HER2 expression levels, with no expression detected in the FMCp cell line. (**B**) After exposure to lapatinib, the CAT-M cell line showed decreased phosphorylation levels of the HER1 (pY1173), HER2 (pY1221 + Y1222) and of its downstream effectors, AKT1 (pS473) and ERK1/2 (pT202/pY204 + pT185/pY187), coupled in an increased of PTEN phosphorylation at Thr366. Results were less perceptible when cells were exposed to neratinib. (**C**) TKi exposure, particularly, lapatinib, increased HER2 levels not only in CAT-M cells, but also in (**D**) FMCp cells. β-actin was used as loading control.

**Figure 3 pharmaceutics-13-00346-f003:**
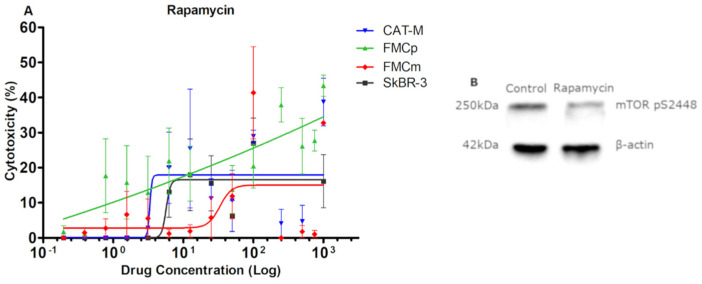
Rapamycin showed small cytotoxicity in all tested tumor cell lines, although it lead to a decrease in the phosphorylation levels of mTOR at S2448. (**A**) The most susceptible cell line to rapamycin exposure was FMCp (43.4% of cytotoxicity) followed by the FMCm (41.3% of cytotoxicity) and the CAT-M cell lines (38.7% of cytotoxicity). The human SkBR-3 cell line was used as control, presenting a maximum of 29.6% of cytotoxicity. The experiments were performed in triplicates and repeated three times. (**B**) Rapamycin reduces the phosphorylation levels of the mTOR at S2448. β-actin was used as loading control.

**Figure 4 pharmaceutics-13-00346-f004:**
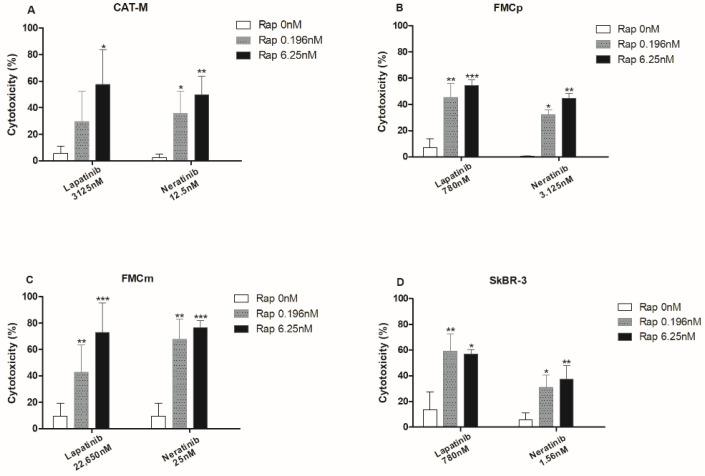
Combinations of TKi and mTOR inhibitor (lapatinib plus rapamycin and neratinib plus rapamycin) showed synergistic antiproliferative effects in feline mammary tumor cell lines. (**A**) CAT-M cells presented a significantly higher antiproliferative response when exposed to lapatinib at 3125 nM plus rapamycin at 6.25 nM or incubated with neratinib at 12.5 nM plus rapamycin at 6.25 nM (57.5% of cytotoxicity, * *p* < 0.05; 49.9% of cytotoxicity, ** *p* < 0.01, respectively). (**B**) FMCp cells showed significant synergistic effects of conjugations between lapatinib at 780 nM plus rapamycin at 0.196 nM or at 6.25 nM (45.5% of cytotoxicity, ** *p* < 0.01; 54.4% of cytotoxicity, *** *p* < 0.001, respectively), and when neratinib at 3.125 nM plus rapamycin at 0.196 nM or at 6.25 nM were used (32.1% of cytotoxicity, * *p* < 0.05; 44.5% of cytotoxicity, ** *p* < 0.01, respectively). (**C**) FMCm cells presented an increased cytotoxic response to the conjugations of lapatinib at 22,650 nM plus rapamycin at 0.196 nM or at 6.25 nM (42.6% of cytotoxicity, ** *p* < 0.01; 95.2% of cytotoxicity, *** *p* < 0.001, respectively) and neratinib at 25 nM plus rapamycin at 0.196 nM or at 6.25 nM (67.7% of cytotoxicity, ** *p* < 0.01; 76.3% of cytotoxicity, *** *p* < 0.001, respectively). (**D**) SkBR-3 cells used as control, displayed similar antiproliferative responses to feline cell lines. All the assays were performed in triplicate and repeated a total of three separated times.

**Figure 5 pharmaceutics-13-00346-f005:**
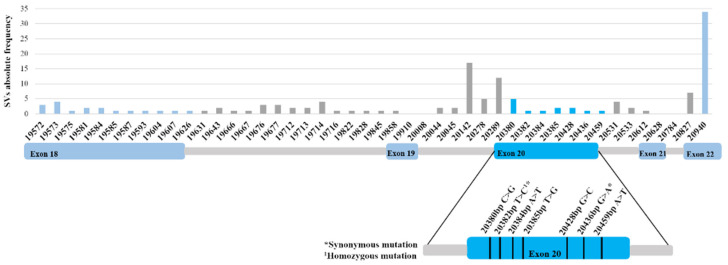
Mutations identified in feline *her2* TK domain are not reported to induce therapeutic resistance in woman breast cancer. Most of the detected mutations were localized at intronic regions, although some of them were identified in exons, as a very frequent mutation in the exon 22 (c.20940 T > G; 85%; 34/40). Regarding the exon 20 that encodes for the protein region recognized by TKi, none of the seven identified mutations were reported as being related to therapeutic resistance in woman breast cancer.

**Figure 6 pharmaceutics-13-00346-f006:**
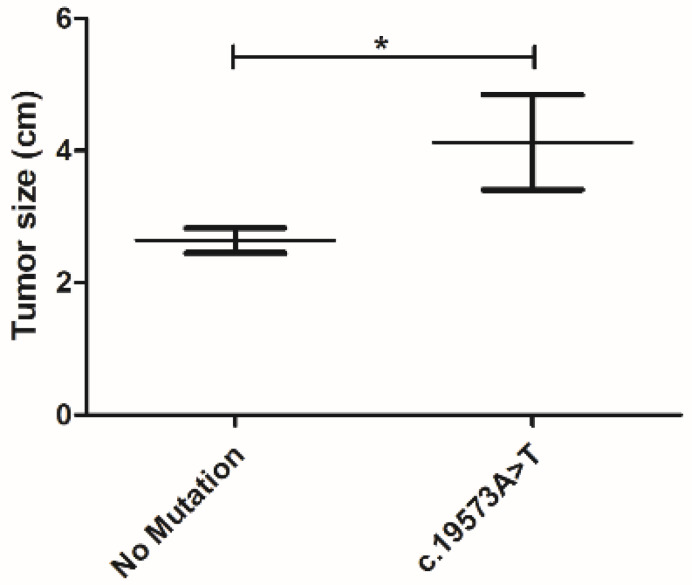
A significant positive correlation was found between the tumor size and the presence of c.19573A > T mutation in exon 18 (* *p* = 0.045). The mutation c.19573A > T was associated to larger tumor sizes (4.13 cm ± 1.44 vs. 2.64 cm ± 1.13).

**Table 1 pharmaceutics-13-00346-t001:** Concentrations of lapatinib (nM), neratinib (nM) and rapamycin (nM), used in the cytotoxicity assays. Cells were exposed to drugs for 72 h before cytotoxicity evaluation.

Drug Concentrations for the Cytotoxicity Assays		
Lapatinib (nM)	Neratinib (nM)	Rapamycin (nM)
195	0.195	0.195
390	0.39	0.39
780	0.78	0.78
1560	1.56	1.56
3125	3.125	3.125
6250	6.25	6.25
12,500	12.5	12.5
25,000	25	25
50,000	50	50
100 × 10^3^	100	100
250 × 10^3^	250	250
500 × 10^3^	500	500
1 × 10^6^	750	750
	1000	1000
	1500	1500

**Table 2 pharmaceutics-13-00346-t002:** Primary antibodies and their dilutions used for chemoluminescence analysis.

Primary Antibody.	Clone	Dilution
β-actin	AC-15	1:5000
HER2	CB11	1:1000
HER2 pY1221 + Y1222	Polyclonal	1:1000
HER1 pY1173	E124	1:500
HER4 pY1284	Polyclonal	1:500
AKT1 pS473	EP2109Y	1:2000
ERK1/2 pT202/pY204 + pT185/pY187	MAPK-YT	1:5000
PTEN pT366	EP229	1:1000
mTOR pS2448	EPR426(2)	1:10,000

**Table 3 pharmaceutics-13-00346-t003:** Clinicopathological features of female cats with mammary carcinomas enrolled in this study (n = 40).

Clinicopathological Feature	Number (%)	Clinicopathological Feature	Number (%)
**Breed**		**Age**	
Indeterminate	33 (82.5%)	<8years old	3 (7.5%)
Siamese	4 (10%)	≥8 years old	37 (92.5%)
Persian	2 (5%)		
Norwegian Forest	1 (2.5%)	**Tumor size**	
		<2cm	9 (22.5%)
		2–3cm	19 (47.5%)
**Spayed**; 1 unknown		>3cm	12 (30%)
Yes	19 (47.5%)		
No	20 (50%)	**HP * classification**	
**Contraceptives**; 7 unknown			
Yes	23 (57.5%)	Tubulopapillary carcinoma	8 (20%)
No	10 (25%)	Solid carcinoma	9 (22.5%)
		Cribiform carcinoma	5 (12.5%)
**Treatment**		Mucinous carcinoma	5 (12.5%)
Mastectomy	36 (90%)	Tubular Carcinoma	11 (27.5%)
Mastectomy + Chemo	4 (10%)	Papillary-cystic carcinoma	2 (5%)
**Multiple tumors**		**HP * Malignancy grade**	
Yes	31 (77.5%)	I	2 (5%)
No	9 (22.5%)	II	5 (12.5%)
**Regional lymph node status**; 2 unknown		III	33 (82.5%)
Positive	14 (35%)	**Tumor necrosis**	
Negative	24 (60%)	Yes	29 (72.5%)
**Stage (TNM classification)**		No	11 (27.5%)
I	9 (22.5%)	**Lymphatic invasion**	
II	7 (17.5%)	Yes	5 (12.5%)
III	21 (52.5%)	No	35 (87.5%)
IV	3 (7.5%)	**Lymphocytic infiltration**	
**Mammary location**		Yes	27 (67.5%)
M1	11 (27.5%)	No	13 (32.5%)
M2	8 (20%)	**Tumor ulceration**	
M3	14 (35%)	Yes	3 (7.5%)
M4	11 (27.5%)	No	37 (92.5%)
**fHER2 status**		**Ki67 index**	
Positive	12 (30%)	Low (<14%)	30 (75%)
Negative	28 (70%)	High (≥14%)	10 (25%)
**ER status**		**PR status**	
Positive	12 (30%)	Positive	20 (50%)
Negative	28 (70%)	Negative	20 (50%)

* HP—Histopathological; TNM—Tumor, Node, Metastasis.

**Table 4 pharmaceutics-13-00346-t004:** Primers for genomic DNA amplification and sequencing, covering the feline *her2* TK domain, between exon 18 to exon 22.

Forward (5′-3’)	Reverse (5′-3´)	Exons
**CTAGTGGAGCCATGCCCAA**	GGAGGTCCCTCCTGTACTCC	18 and 19
**AATCTTGGACGTAAGCCCCTC**	AGGCCCCCTAAGTGCATACC	20
**CTGACATCCACCGTGCAGTT**	CGTAGCTCCACACGTCACTC	21 and 22

## Data Availability

The datasets used and analyzed in the current study are available from the corresponding author in response to reasonable requests.

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
