# Peer review of "Tyrosine Kinase Inhibitors Are Promising Therapeutic Tools for Cats with HER2-Positive Mammary Carcinoma"

_pharmaceutics, 2021, doi:10.3390/pharmaceutics13030346_

Round 1

Reviewer 1 Report

Although the paper provides promising outcomes, I have some concerns and comments that need to be clarified and addressed before further consideration.

Major points

Materials and Methods
2.1. Feline and human mammary carcinoma cell lines
Did the authors use human carcinoma cell lines? If so, why did the author stated in the Conclusion section that ''This study emphasizes the valuable use of the cat as a human breast cancer model...''. Please clarify this point and please specify all used materials correctly. This point should be applied to all materials, which are used in the study.

2.2. In vitro cytotoxicity assays
Table 1. why Lapatinib is presented in (μM) concentration, while Neratinib and Rapamycin are presented in (nM) concentrations? Please clarify.

2.3. Immunoblotting
Please provide a reference for the used immunoblotting methodology. This point important to ensure that the procedure has been conducted properly according to a verified method. 

2.5. DNA extraction, amplification, and sequence analysis of the feline her2 TK domain
Please provide a reference for the used methodologies.
Based on which criteria the used primers were chosen? Please clarify and address this point in the text. 

Results
Figures 1 and 3. Are the authors sure that the presented logs of drug concentrations are presented correctly? Please check this point and correct the probable errors. 

3.5. Mutations found in the her2 TK domain of FMC clinical samples were not associated with 292TKi therapy resistance
I am not sure if the used method for determining the mutations was enough to build on and to claim data presented in this section. Please clarify this point and add a reasonable explanation in the text.  
Figure 5. Please replace this figure with a sharper image (high resolution). 

Discussion
This section needs major improvement. The results are poorly discussed, where the results should be more rationalized with previously reported studies. I recommend the authors add more inputs to this section.  

Conclusions
Lines 425-427. The statement ''Altogether, our data strongly demonstrated the usefulness of TKi alone or combined with rapamycin in the treatment of cats with feline mammary carcinoma, since mutations found suggest that 426TKi-resistant FMC are very rare'' should be reworded, where the obtained data showed potential effects as the experiments were performed at the level of in vitro. Therefore, the obtained results have potentially beneficial effects.  

Minor points
Line 194. IC50, 50 should be written in a lower index. Please check the full text for such errors.
I recommend the authors check the full text for grammatical and typing errors.

Author Response

Reviewer 1:

Dear reviewer, thank you so much for your corrections and remarks that improved the final quality of our work. We hope that after the performed corrections, you may consider this manuscript suitable for publication.

Major points

Materials and Methods

“2.1. Feline and human mammary carcinoma cell lines

Did the authors use human carcinoma cell lines? If so, why did the author stated in the Conclusion section that ''This study emphasizes the valuable use of the cat as a human breast cancer model...''. Please clarify this point and please specify all used materials correctly. This point should be applied to all materials, which are used in the study.”

Dear reviewer, indeed we used a human HER2-overexpressing breast cancer cell line (SkBR-3) as a positive control for the tested compounds. As suggested, the use of this cell line was better clarified in M&M section (line 94). Regarding the statement of the authors in the Conclusion section, we intended to highlight that the three used feline cell lines were sensitive to TKi and rapamycin through the same mechanisms of action already reported for the human SkBR-3 cell line, suggesting that feline cell lines can be used as in vitro tools for the screening of new drugs in both species. This consideration was added to the Conclusions section (lines 479 to 485).

“2.2. In vitro cytotoxicity assays

Table 1. why Lapatinib is presented in (μM) concentration, while Neratinib and Rapamycin are presented in (nM) concentrations? Please clarify.”

Dear reviewer, thank you for this question. Indeed, we used different concentration units, but this could be confusing for the readers. As also recommended by Reviewer 2, and in order to simplify, lapatinib concentration was converted to nM along the manuscript. Text and figures were revised accordingly.

 “2.3. Immunoblotting

Please provide a reference for the used immunoblotting methodology. This point important to ensure that the procedure has been conducted properly according to a verified method.”

Dear reviewer, the immunoblotting assay was performed considering methodology already described by our group, using feline cell lines, and also as reported by other groups using human cell lines (references 2, 26, 27 and 28; lines 123 to 124).

 “2.5. DNA extraction, amplification, and sequence analysis of the feline her2 TK domain

Please provide a reference for the used methodologies.

Based on which criteria the used primers were chosen? Please clarify and address this point in the text.”

Dear reviewer, as suggested the description of the methodology was improved and some references were added (references 31, 32 and 33; lines 161 to 164).

Regarding the primers’ design, we used the Primer Designing Tool from NCBI, blasting the feline her2 sequence (NC_018736.3) with the annotated human her2 sequence (NC_000017.11). This explanation was added to the section (lines 169 to 173 and 188 to 192). In addition, an improved version of the methodology towards the identification of the mutations that putatively lead to TKi resistance was added (lines 195 to 199).

 Results

“Figures 1 and 3. Are the authors sure that the presented logs of drug concentrations are presented correctly? Please check this point and correct the probable errors.”

Dear reviewer, thank you for this important correction. Indeed, although the concentrations of the drugs were represented correctly, the linear scale was not properly defined. Thus, new figures 1 and 3 were uploaded with the XX axis corrected.

 “3.5. Mutations found in the her2 TK domain of FMC clinical samples were not associated with TKi therapy resistance

I am not sure if the used method for determining the mutations was enough to build on and to claim data presented in this section. Please clarify this point and add a reasonable explanation in the text. 

Figure 5. Please replace this figure with a sharper image (high resolution).”

Dear reviewer, the mutations found in the feline tumor tissue samples were compared with the data related to the human her2 sequence, as clarified in the Materials and Methods. Afterward, the identified mutations were searched in three databases (National Cancer Institute database, International Cancer Genome Consortium, and Catalogue of Somatic Mutations in Cancer) in order to identify if they are reported as inducing resistance to therapy. The name of the used databases was added to the Results section (lines 313 to 314) as well as a brief explanation on how we defined if the mutations found could lead to therapy resistance (lines 333 to 335).

As also suggested by you, the definition of Figure 5 was improved and changed for a 300dpi resolution image.

Discussion

“This section needs major improvement. The results are poorly discussed, where the results should be more rationalized with previously reported studies. I recommend the authors add more inputs to this section.”

The discussion section was revised and improved accordingly to the suggestions. The obtained results were further discussed with other studies (10 new references were added). Thank you so much for this important request that certainly will improve the final quality of the manuscript.

Conclusions

“Lines 425-427. The statement ''Altogether, our data strongly demonstrated the usefulness of TKi alone or combined with rapamycin in the treatment of cats with feline mammary carcinoma, since mutations found suggest that TKi-resistant FMC are very rare'' should be reworded, where the obtained data showed potential effects as the experiments were performed at the level of in vitro. Therefore, the obtained results have potentially beneficial effects.”

Dear reviewer, thank you for this correction. The sentence was revised as you suggest becoming more accurate (lines 491 to 494).

 Minor points

“Line 194. IC50, 50 should be written in a lower index. Please check the full text for such errors.

I recommend the authors check the full text for grammatical and typing errors.”

Dear reviewer, thank you for this correction and recommendation. All the manuscript was revised by a native English speaker in order to avoid typing errors and to improve the text.

Reviewer 2 Report

The manuscript “Tyrosine kinase inhibitors are promising therapeutic tools for cats with HER2-positive mammary carcinoma”. It is a very interesting work describing the antiproliferative effects of two TKi (lapatinib and neratinib), and their combination with a mTOR inhibitor (rapamycin). I recommend this manuscript to be published in Pharmaceutics with a few recommendations.

Line 99. “CO2” should be CO2

Table 1. Why the authors are using different units in the concentration of the drugs, they must place them in the same units because this only causes confusion for the readers. This occurs throughout the manuscript.

Line 267, 264. “IC50” should be IC50

Author Response

Reviewer 2:

Dear reviewer, thank you so much for your review and comments in order to improve the quality of our manuscript, and also for giving your positive remarks on our work.

“Line 99. “CO2” should be CO2

Changed as requested (line 100).

“Table 1. Why the authors are using different units in the concentration of the drugs, they must place them in the same units because this only causes confusion for the readers. This occurs throughout the manuscript.”

Dear reviewer, thank you for this highlight. You are completely right and all the concentrations were changed for the same units, along with the manuscript.

“Line 267, 264. “IC50” should be IC50

As requested, all the “IC50” abbreviations were corrected along with the manuscript (lines 126, 205, 215 to 221, 226 to 231, 369 to 372, and 378).

Round 2

Reviewer 1 Report

All concerns and comments have been addressed and clarified, and the paper has been significantly improved.